# Bacterial Growth on Three Non-Resorbable Polytetrafluoroethylene (PTFE) Membranes—An In Vitro Study

**DOI:** 10.3390/ma15165705

**Published:** 2022-08-18

**Authors:** Helena Zelikman, Gil Slutzkey, Ofir Rosner, Shifra Levartovsky, Shlomo Matalon, Ilan Beitlitum

**Affiliations:** 1Department of Oral Rehabilitation, Goldschleger School of Dental Medicine, Sackler Medical Faculty, Tel Aviv University, Tel Aviv 6997801, Israel; 2Department of Periodontology and Dental Implantology, Goldschleger School of Dental Medicine, Sackler Medical Faculty, Tel Aviv University, Tel Aviv 6997801, Israel

**Keywords:** barrier membranes, GBR, membrane exposure, bacterial contamination

## Abstract

GBR (Guided Bone Regeneration) procedure is challenged by the risk of membrane exposure to the oral cavity and contamination. The barrier quality of these membranes serve as a mechanical block from bacterial penetration into the GBR site. The purpose of this in vitro study was to evaluate the antibacterial effect of three commercial non-resorbable polytetrafluoroethylene membranes. (Two d-PTFE membranes and one double layer e-PTFE +d-PTFE membrane). A validated in vitro model with two bacterial species (*Streptococcus sanguinis* and *Fusobacterium nucleatum*) was used. Eight samples from membrane each were placed in a 96-well microtiter plate. The experimental and positive control groups were exposed to a bacterial suspension which involved one bacterial species in each plate. Bacterial growth was monitored spectrophotometrically at 650 nm for 24 h in temperature controlled microplate spectrophotometer under anaerobic conditions. One- Sample Kolmogorov–Smirnov Normal test and the Kruskal–Wallis test was used for the statistical analysis. As shown by the bacterial growth curves obtained from the spectrophotometer readings, all three membranes resulted in bacterial growth. We have not found a statistical difference in *F. nucleatum* growth between different membrane samples and the positive control group. However, *S. sanguinis* growth was reduced significantly in the presence of two membranes (CYTOPLAST TXT-200 and NeoGen^TM^) when compared to the control (*p* < 0.01). The presence of Permamem^®^ had no significant influence on *S. sanguinis* growth. Some types of commercial non-resorbable PTFE membranes may have an impact on the growth dynamics of specific bacterial species.

## 1. Introduction

Guided Bone Regeneration (GBR) has been proved to be safe and predictable in managing maxillary and mandibular ridge deficiencies associated with dental implant placement [1,2]. The GBR technique is based on the principle of compartmentalization, achieved by a surgical placement of cell occlusive membrane facing the bone surface in the site of the osseous defect. The membrane creates and maintains a secluded space for the migration and proliferation of pluripotent and osteogenic cells by the exclusion of epithelial and connective tissue cells from the osseous defect [1,2,3]. The technique is challenged by possible membrane exposure that leads to bacterial colonization of the membrane [4,5]. This undesirable sequence of events may compromise the GBR outcomes and lead to a failure of the procedure [6,7,8,9,10,11,12].

Expanded polytetrafluoroethylene (e-PTFE) membranes were the first available commercial non-resorbable membranes introduced on the market [12,13]. PTFE is a highly stable polymer, biologically and chemically inert and is very resistant to enzymatic attacks [14]. These membranes were frequently used in GBR procedures and provided predictable outcomes [15]. A common feature of variable synthetic membranes is a porosity ranging between <8 μm and 300 μm. This property is believed to facilitate tissue cell attachment, promotion of wound and membrane stability and enable the diffusion of nutrients through the membrane to stimulate new bone formation during the early healing phase [16,17,18]. E-PTFE membranes are composed of two layers; an inner layer with open microstructure facing the bone and a thick outer layer, the occlusive structure facing the soft tissues. Previous studies concluded that the porous part of the membrane aids bacterial biofilm formation, proliferation and subsequent bacterial penetration from the outer to the inner portion of the membrane within 4 weeks [5,16,17,18,19]. Thus, e-PTFE membranes require efficient soft tissue coverage or primary closure of the flap, in order to avoid membrane exposure and bacterial contamination [19,20,21,22].

The high density PTFE membranes are the alternative to e-PTFE membranes with the same indications, but different properties [16,17,18,19,20,23,24,25,26,27]. The d-PTFE membranes have a denser structure compared to e-PTFE membranes, they do not expand and have lower porosity (0.2 μm). Thus, it has been suggested that d- PTFE membranes are more resistant to bacterial penetration [13,16,26]. However, the low porosity of d-PTFE membranes can limit blood supply in the surgical area. Moreover, these membranes, do not facilitate cell adhesion to their surface, hence simplifying their removal through the mucosal flap during the second surgery. According to several publications, d-PTFE membranes do not necessitate primary closure and do not require immediate removal in the case of exposure [28,29,30,31,32,33].

A previous in vitro study from this research group evaluated bacterial growth on resorbable collagen membranes. It was found that one of the membranes accelerated the bacterial growth rate. It was concluded that the clinician has to be careful when using certain resorbable membranes [34]. The study was based on the direct contact test (DCT) technique, which allows measuring the effect of close and direct contact between the bacteria and the tested material on the bacterial viability, independent of the solubility and diffusability of the material tested [35].

The aim of this in vitro study was to evaluate the bacterial growth on three commercial non-resorbable polytetrafluoroethylene (PTFE) membranes using the DCT technique.

## 2. Materials and Methods

### 2.1. The Tested Membranes

Three commercial non-resorbable membranes were tested:Permamem^®^, a high density PTFE membrane (Botiss biomaterials GmbH, Zossen, Germany).CYTOPLAST TXT-200, a high density PTFE membrane (Osteogenics Biomedical, Inc., Lubbock, TX, USA). The membrane has a closed microstructure with hexagonal indentations [36].NeoGen^TM^, a double layer n- PTFE membrane: the inner layer (facing the hard tissue) is an e-PTFE and the outer layer (facing the soft tissue) is a tight PTFE. (Neoss LTD., Harrogate, UK). The membrane has an open microstructure with multidirectional polymer fibril orientation [36].

### 2.2. Material Characterization

Square specimens (6 × 6 mm^2^) were cut from polytetrafluoroethylene (PTFE) membranes under sterile conditions. One sample of each membrane was mounted on aluminum stubs using conductive carbon tape. Scanning electron microscopy was performed in a Zeiss GeminiSEM 300—HRSEMA (Carl Zeiss QEC GmbH, Peine, Germany) in order to assess the surface topography of the membranes. The images were taken at a low vacuum mode in a pure Nitrogen atmosphere of ~100Pa with an accelerating voltage of 3 kV.

Static contact angle measurements were performed using a Rame-Hart model 400 Goniometer and The DROPimage Pro software (Ramé-hart instrument co. Succasunna, NJ, USA). Droplets of water were used for surface wettability analysis.

The surface roughness of one sample of each membrane was estimated with a confocal laser scanning microscope (CLSM) Olympus OLS 4000 LEXT (Olympus Corporation, Tokyo, Japan) with a cut-off value set at 0.08 mm.

### 2.3. The Tested Bacteria

The tested bacteria were *Streptococcus sanguinis* (SSs34), an anaerobic facultative bacteria, known as an early colonizer and a frequent isolate from dental biofilm, associated with healthy dental implants [37,38,39,40] and *Fusobacterium nucleatum* (Fn1594), an anaerobic, gram-negative bacteria, known as one of the periodontal and periimplantitis pathogens [40,41,42,43,44,45,46,47,48,49,50,51].

### 2.4. Study Model

The study was based on a direct contact test (DCT) technique. We used and reported that particular methodology previously [34,35]. The bacterial growth was recorded continuously on a 96-well microtiter plate (96-wells flat bottom Nuclon, Nunc) in a temperature controlled spectrophotometer set to 37 °C (VesaMax, Molecular Device Corporation, San Jose, CA, USA). The device was set to measure the optical density at 650 nm every 30 min for 24 h. To ensure homogenous bacterial suspension prior to each reading the device was set to perform auto-mixing.

Eight samples of each tested membrane sized 4 × 10 mm were fixed to special polycarbonate inserts inside the wells. Then the plate was held vertically and 10 microliter (approximately 10^6^ cells) of bacterial suspension was dripped on each membrane sample in the tested group (Groups A, C, E). After this phase, the plate was inserted into an incubator for 40 min at 37 °C, to assure fluid evaporation from the bacterial suspension and to establish close and direct contact between the bacteria and the membranes. Later, 235 mL of the relevant medium (HBI broth for the *S. sanguinis* and Wilkins–Chalgren anaerobic agar for *F. nucleatum*) was added to the wells in the tested groups and the plate was gently vibrated for 2 min (Gyrotory Shaker, New Brunswick Scientific, Enfield, CT, USA). Then 15 mL of suspension were transferred from the wells with the membranes (Groups A, C, E) to the wells containing 205 mL of sterile growth medium (Groups B, D, H). In the result, two sets of eight wells (with and without the membrane) containing an equal volume of relevant liquid medium (220 mL), made it possible to monitor bacterial growth in the presence and without the presence of the tested membranes.

One column of wells with the membrane samples covered with an equal volume of uninoculated fresh medium served as the negative control group.

Bacterial growth was monitored spectrophotometrically at 650 nm for 24 h in a temperature-controlled microplate spectrophotometer under anaerobic conditions (Thermomax, molecular devices corporation, Menlo park corporate center, Menlo Park, CA, USA). The kinetics of bacterial growth from all the experimental groups were recorded in OD (optical density) units and the growth curves were compared to the control outgrowth. (Figure 1) The study model.

### 2.5. Calibration Test

The calibration test was performed simultaneously, to monitor bacterial outgrowth under the same experimental conditions, in a quantitative and standardized manner. Ten microliters of bacterial inoculum were placed on the side wall of three wells as in the experimental design; 275 mL of fresh growth medium were added into each well, and the contents of the wells were mixed for 2 min. From each of the three wells, 55 mL of the medium were transferred to the next set of wells that contained 220 mL of sterile medium. The dilution transfer was repeated five consecutive times. The first row of wells in this group of wells served also as a positive control group.

The negative control OD values were considered the baseline and were subtracted from the experimental data. The growth curves for each well were plotted, and the regression line was calculated on the ascending linear portion of the curve. As a result of this analysis two parameters were calculated, the bacterial growth rate and the number of bacteria initially.

The statistical analyses were performed using the one-Sample Kolmogorov–Smirnov Normal Test to find out if the observed cumulative variables have a normal distribution. Then the Kruskal–Wallis test was submitted for both parameters.

## 3. Results

### 3.1. Material Characterization

The results are summarized in the table (Figure 2 and Figure 3).

The NeoGen^TM^ membrane exhibited values of static contact angles, followed by Permamem^®^. Cytoplast TXT-200 was the least hydrophobic of the three tested membranes. Permamem^®^ showed the greatest surface roughness, followed by NeoGen^TM^. Cytoplast TXT-200 had the lowest surface roughness.

### 3.2. Calibration Test

The calibration test showed a gradual decline in the amount of bacteria as a result of five consecutive dilutions at the time 0 and made little impact on the density of the bacteria in the stationary phase in this system. Under the experimental conditions, a 10⁵ reduction in the number of living bacteria at time 0, caused approximately 1 h delay in the exponential phase of the curve. According to these results, we can establish the antibacterial activity of the tested membranes as a reduction in bacteria numbers in the DCT system. (Figure 4).

### 3.3. Bacterial Growth in the Presence of the Membranes

#### 3.3.1. *Fusobacterium nucleatum*

The presence of the three tested membranes samples: Permamem^®^, CYTOPLAST TXT-200 and NeoGen™ did not demonstrate a statistical difference (*p* < 0.312) in *F. nucleatum* growth rate in comparison to the bacterial growth in the control group. (Figure 5 and Figure 6).

#### 3.3.2. *Streptococcus sanguinis*

The growth rate of *S. sanguinis* was significantly retarded by the presence of the membranes NeoGen^TM^ (*p* < 0.001) and CYTOPLAST TXT-200 (*p* < 0.008) when compared to the control group. However, the presence of Permamem^®^ membrane did not disturb the growth of the bacteria as compared to the control group. Moreover, we found a statistical difference between the rates of bacterial growth in the presence of Permamem^®^ membrane as compared to the bacterial growth rate in the presence of NeoGen^TM^ membrane (*p* < 0.004). (Figure 7 and Figure 8).

## 4. Discussion

In the present study, we examined the antibacterial activity of three different commercial non-resorbable membranes in the presence of two bacterial species, *S. sanguinis* and *F. nucleatum*. Two non-resorbable membranes, double layer PTFE membrane (NeoGen^TM^, Neoss LTD. Harrogate, UK) composed of tight PTFE and e-PTFE layers and a d-PTFE membrane (CYTOPLAST TXT-200, Osteogenics Biomedical, Inc., Lubbock, TX, US) have shown the ability to hinder *S. sanguinis* (an early colonizer species) growth compared to another d-PTFE membrane (Permamem^®^, Botiss biomaterials GmbH, Zossen, Germany). Nonetheless, we found no significant difference in the *S. sanguinis* growth rate in the presence of d-PTFE membrane (CYTOPLAST TXT-200) as compared to the growth rate in the presence of double layer e-PTFE membrane (NeoGen^TM^). The results of our study also show that the presence of the tested membranes had no antibacterial effect on the growth of *F. nucleatum*.

Being an early colonizer, *S. sanguinis* inhibits every surface inside the oral cavity, binding other periopathogens, such as *Porphyromonas gingivalis* and *Fusobacterium nucleatum*. *F. nucleatum* is a periodontal pathogen associated with a wide array of human diseases involving chronic and aggressive periodontitis [51]. It has been reported to coaggregate with many oral microorganisms, such as *S. sanguinis*, *S. mutans* and *P. gingivalis* [52]. Moreover, it enhances the coaggregation between *S. sanguinis* and *P. gingivalis* [52].

The above results are compatible with previous reports. Trobos et al. did not find a correlation between membrane characteristics (wettability, surface roughness) and the adhesion and growth of *S. oralis* (early colonizer bacteria) on tested d-PTFE and e-PTFE membranes [36]. In clinical research, Turri et al. [53] observed a distinct pattern of bacterial colonization on different types of non-resorbable membranes in an oral environment. E-PTFE membranes have shown less bacterial adhesion and proliferation as compared to the d-PTFE membranes, similar to the results of another in vitro study conducted by Trobos with *S. oralis* species, known as early colonizers [36].

We found no correlation between membrane characteristics, such as wettability and bacterial growth rate. NeoGen^TM^ was the most hydrophobic membrane and Cytoplast TXT-200 was the least hydrophobic membrane tested. However, both of the membranes, the e-PTFE and the d-PTFE type, had a negative influence on the bacterial growth rate. Since *S. sanguinis* is an early colonizer, they may show a higher affinity to rougher membranes, such as Permamem^®^, which was found to be the roughest of all three membranes. On the other hand, *F. nucleatum* relies on early colonizers, such as *S. sanguinis* to form the biofilm, thus *F. nucleatum* alone grew at the same rate near all three tested membranes.

It has been hypothesized that membrane microtopographic patterns and surface roughness have an impact on the phase of biofilm attachment and proliferation [54,55]. Different topographic features, such as microfibrils of the e-PTFE membranes or lines or its on the d-PTFE membranes serve as a platform promoting microbial adhesion and aid in biofilm proliferation [36,53]. Hoe et al., have found that thicker microbial biofilms are produced inside 400 μm pits as compared to smooth surfaces [56]. However, surface roughness has a variable influence on the adhesion and biofilm formation proliferation among different bacterial species and environmental factors [36].

In the oral environment, bacterial adhesion depends not only on surface topography but also on the adhesion of the acquired pellicle (proteins) to the substrate and the surface free energy. For instance, surfaces with more hydrophobic characteristics, such as PTFE membranes have better adsorption of proteins. Previous studies have found positive correlation between biofilm formation and the surface energy of various materials. Trobos et al., have found that e-PTFE and d-PTFE membranes have similar surface energy [36]. This finding can partially explain comparable plaque accumulation on different types of membranes.

A number of previous studies examined the impact of membrane exposure and bacterial colonization and penetration of the barrier membrane. Membrane exposure has a detrimental effect on regenerative procedure outcomes and the bacterial contamination may lead to the penetration of bacteria followed by fibroblasts and giant cells [10,11,26,27]. Mombelli et al. [37,38] found that e-PTFE membranes used for periodontal regeneration were frequently colonized by periopathogens. Wang et al. [7] found selective bacterial adhesion patterns to different types of barrier membranes.

*Streptococcus sanguinis* is a commensal, anaerobic facultative, gram-positive cocci. This microbe is part of the oral flora, known as an early colonizer, frequently isolated from bacterial biofilms and healthy implant sites [39,40,41,42,43,44,45,46]. This bacteria species can adhere to dental surfaces, oral mucosa and survive in saliva. The microbial cell is equipped with fimbriae and adhesins to assist the bacterial attachment to various surfaces. Moreover, this bacteria possesses external appendages with many hydrophobic domains consisting of non-polar hydrophobic acids, which increases the probability of hydrophobic interactions with other cells and surfaces [39,40,41,42,43,44,45,46,47]. *S. sanguinis* has been shown to be more hydrophobic compared to other early colonizers, such as *S. mitis* and *Actinomiyces* sp. [42].

The role of this bacterium in periodontal disease is yet to be established, but it was speculated that S.sanguinis is a target for periopathogen attachment, such as *P. gingivalis* and *F. nucleatum* [41,42,43,44,45,46,47].

*Fusobacterium nucleatum* is an anaerobic gram-negative rod, a known and established periopathogen [46,47]. Recent studies reported that *F. nucleatum* is one of the first massively increased species during the progression of mucositis to peri-implantitis [48]. This bacteria has the ability to attach to early colonizers cells (gram-positive cocci, *S. sanguinis*, for example) inside the dental plaque and plays a role in the attachment of other periopathogens (*P. gingivalis*, etc.). It serves as a bridging organism inside a mature plaque, and adheres to both early and late colonizers, promoting the coaggregation of periodontal disease-related pathogens [46,47,48,49,50,51,52]). Thus, membrane surface characteristics may have a minor impact on the growth of *F. nucleatum*.

The present study has its limitations. First and foremost, it is a preliminary in vitro study. We tested the antibacterial properties of the membranes on only two bacterial species, which represent periodontal and peri-implant disease and healthy microbiota. The mediums used in our study were commercially available growth mediums and not human saliva or saliva substitute which may alter the surface characteristics of the tested membranes and as a result, bacterial adhesion and proliferation [57]. Thus, the study model can hardly mimic the oral environment and the diversity of oral microflora. Furthermore, this study model enables only short observations on bacterial growth dynamics, since the bacteria growth curves reach the plateau within 7–12 h. The results of a longer study model may be different. Therefore, more in vitro and clinical research is needed in the field for a better understanding of non-resorbable membrane antibacterial properties.

## 5. Conclusions

The results of this preliminary study have shown different membranes may have different influences on the growth rate of *S. sanguinis*. This finding may have a clinical impact, especially if the membrane is exposed to the oral cavity during the healing phase, since particular membranes may inhibit early colonizers, such as *S. sanguinis* and thus may hinder biofilm formation during the healing phase. Moreover, although all tested bacteria were in contact with d-PTFE membranes, the membrane with the highest roughness values showed the highest bacterial growth rate with the early colonizing bacteria. More studies testing other properties and characteristics of commercial membranes are needed, primarily to investigate the correlation between membrane properties and bacterial growth on the molecular level.

## Figures and Tables

**Figure 1 materials-15-05705-f001:**
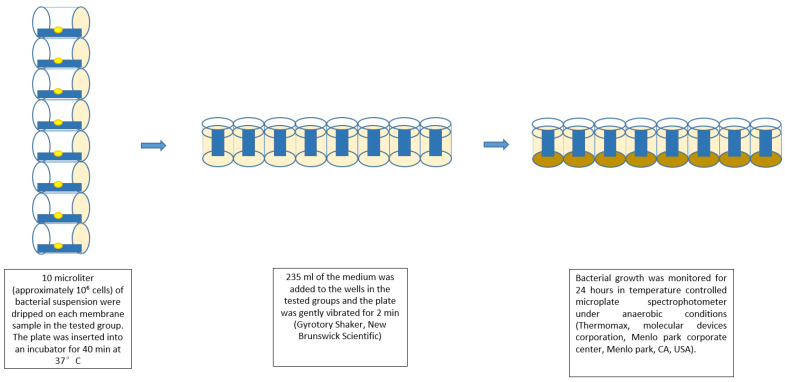
The study model workflow.

**Figure 2 materials-15-05705-f002:**
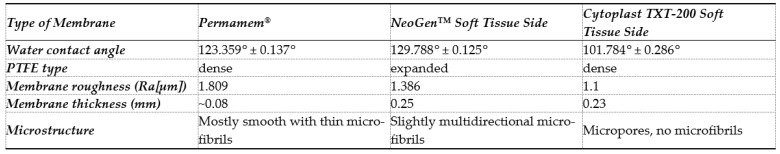
Material characterization of non-resorbable polytetrafluoroethylene (PTFE) membranes.

**Figure 3 materials-15-05705-f003:**
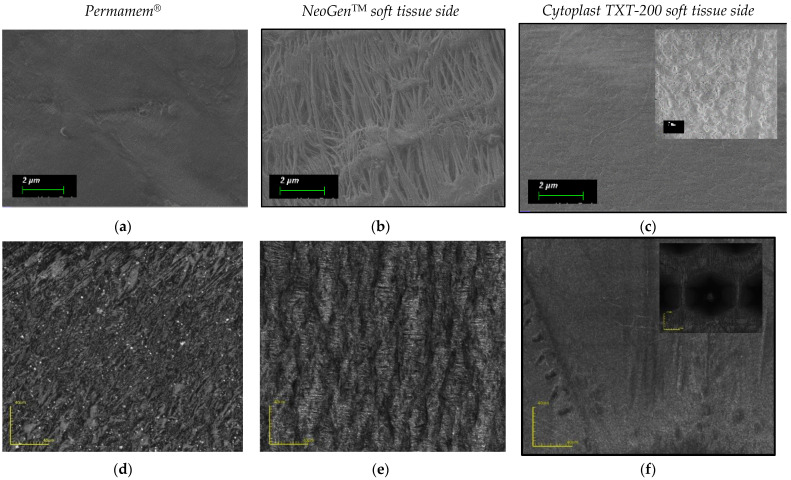
Nonresorbable polytetrafluoroethylene (PTFE) membranes SEM and CLSM images. Bar scale–2 µm. (**a**–**c**) Please note the microfibril microstructure of the e-PTFE membrane (NeoGen^TM^). (**b**) Bigger magnification reveals the micropore structure of the Cytoplast TX-200 membrane. (Bar scale-1 µm). (**c**) Images (**d**–**f**) performed by confocal microscope (CLSM) show the microtopography of the tested membranes (Bar scale 40 µm). Hexagonal indentations on the Cytoplast TXT-200 membrane surface (**f**) in the 200 µm bar scale.

**Figure 4 materials-15-05705-f004:**
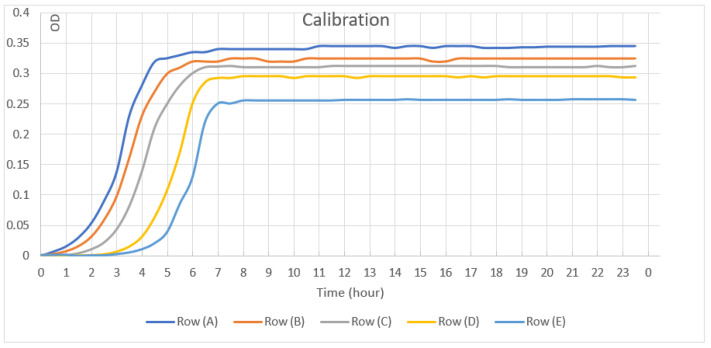
Calibration test results. Bacterial growth after 5 consecutive dilution transfers.

**Figure 5 materials-15-05705-f005:**
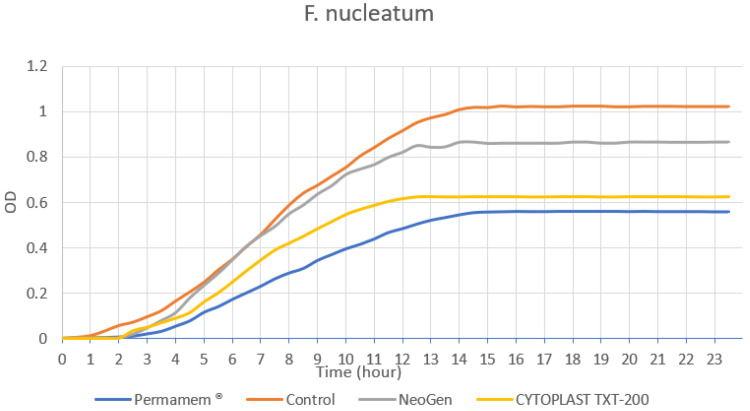
*F. nucleatum* growth curves.

**Figure 6 materials-15-05705-f006:**
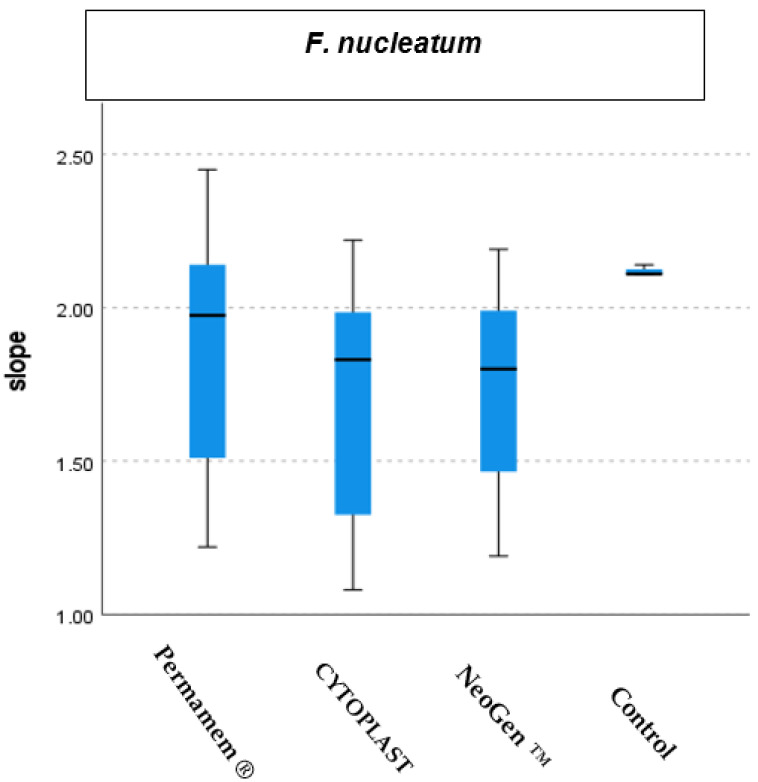
*F. nucleatum* growth distribution in the presence of the tested membranes and in a control group.

**Figure 7 materials-15-05705-f007:**
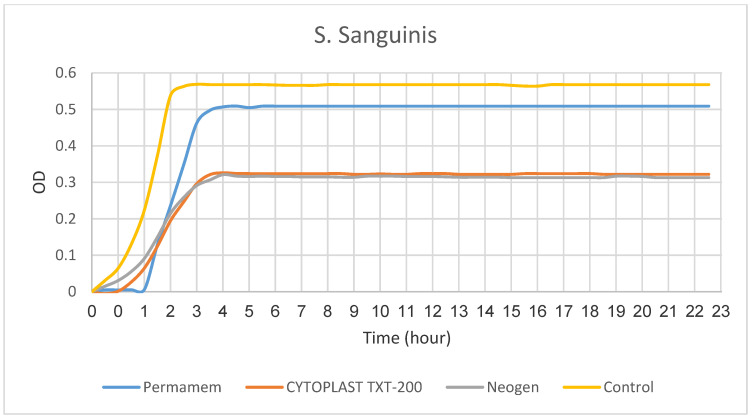
S.sanguinis growth rate.

**Figure 8 materials-15-05705-f008:**
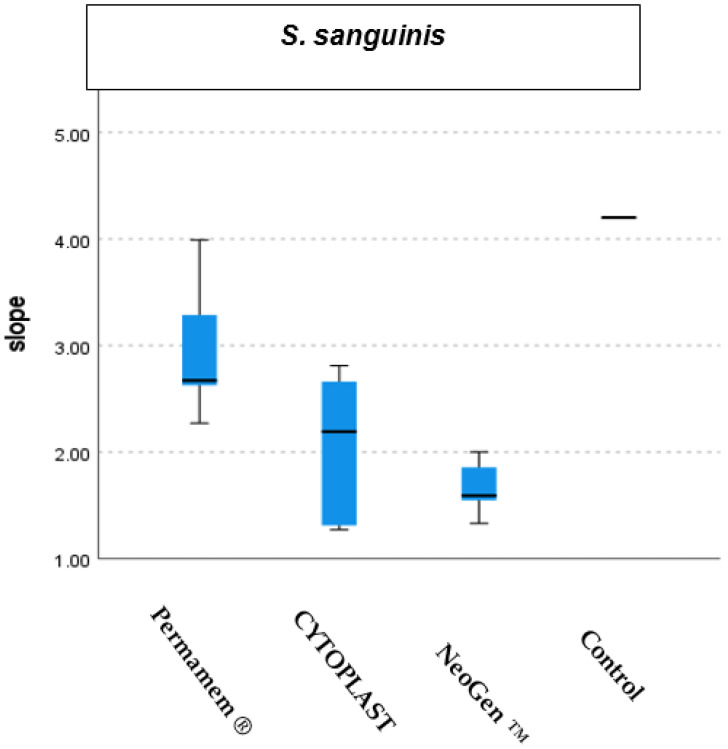
*S. sanguinis* growth distribution in the presence of the tested membranes and in a control group.

## Data Availability

Not applicable.

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
