# Peer review of "Bacterial Growth on Three Non-Resorbable Polytetrafluoroethylene (PTFE) Membranes—An In Vitro Study"

_materials, 2022, doi:10.3390/ma15165705_

Round 1

Reviewer 1 Report

Dear authors, 

The study entitled "Bacterial growth on three non-resorbable polytetrafluoroethylene (PTFE) membranes. An in vitro study" reports a simple study with 3 different commercial types of membranes for bone regeneration and their contaminations with 2 bacteria strains. The study is interesting for clinicians because apply 3 commercial membranes and the topic of infections associated with membranes is current. However, the study is quite simple and short. The study has a huge lack regarding the most important factor of bacterial contamination on membranes which is the superficial characteristics of each membrane. The authors discussed it in a few sentences but no analysis was demonstrated. Moreover, the discussion about these superficial properties needs extensive improvement. 

Some major issues should be improved to achieve merit for publication: 

1- Membrane characterization: The authors must show the superficial characteristics of each brand in this study and the chemical composition of each membrane. Morphology, roughness, chemical composition, and changes in the superficial features, are the main points to have different levels of bacteria proliferation and biofilm formation. Even same, the membranes are commercial the authors should show the differences between them. Only the brand name and description are not enough. The characterization should appear in Materials and Methods and results about this in the Results section.

2- In-deep discussion about the impact of superficial properties on bacterial adhesion and proliferation should be addressed. Again, this is the main point because the authors are showing differences in the results. 

3- Limitation: The study should describe the limitation that the authors investigated only until 24 hours. After this time, the levels of bacteria may be equal or not. 

4- The conclusion of the study is vague. The authors didn't explain why the results were different or significant. But the authors suggest that the study might be important for the clinical choice of membranes. This point returns to the question of superficial characterization, with the information on the surface properties the authors can create appropriate conclusions and suggestions. 

5- Intense literature and citations are missing in this study due to the non-investigation of material properties. Additionally, the results can be totally different in in vivo models, and clinical studies, also after contamination with saliva or bacterias and literature about that need to be addressed. Below I suggest the reading of literature that supports all my criticism of my comments: 

- DOI: 10.1016/j.biomaterials.2016.01.034

- DOI: 10.3390/ijms23042024

Author Response

Dear Reviewer, 

Thank you for your time and efforts to revise our manuscript. The comments were revised one by one and certainly improved the quality of our manuscript. Below, you can find the corrections comment by comment. 

  1. We conducted an additional research that included membrane surface characterization with SEM, static contact angle measurements and surface roughness evaluation. We added the membrane characterization in the "Materials and Methods" and the "Results" sections. Fig 2 and 3 were added.
  2. The discussion section was corrected to clarify and to provide more information about the impact of the membrane superficial properties .228-235
  3. Thank you for your remark regarding the limitation of our study and we elaborated of that in lines 276-285
  4. The conclusion paragraph was changed  according to your remark (lines 288-296.)
  5. Thank you very much for the references you attached, we added more references accordingly.
  6. Best Regards, hope to hear from you soon

Reviewer 2 Report

There are some suggestions-

Abstract

The abstract lacks scientific basis, such as the difference and antibacterial mechanism of three commercially available PTFE and carrier antibacterial.

Introduciton-

The same problem arises in this section, the lack of scientific background on the application of PTFE and the treatment of bacteria by PTFE.

Section 2.1.

Detailed composition or material differences should be shown here.

Results

This manuscript focuses on the direct contact test model to measure the ability of three commercial PTFEs against bacteria and, therefore, significance should be addressed.

Author Response

Dear Reviewer, 

Thank you for your remarks regarding our manuscript entitled “Bacterial growth on three non-resorbable polytetrafluoroethylene (PTFE) membranes. An in vitro study”.

  1. Thank you for your comment, we added in the abstract section the scientific basis for the barrier membrane action although in short since this is the abstract section. (lines 12-15)
  2. The introduction now contains further remarks about the antibacterial influence of the PRFE membranes.
  3. Section 2.2 according to your remark additional research took place with results elaborated including surface characteristics and photomicrographs. (lines 88-99, 160-176) 
  4. In the results section, significance was added line 178 -183.
  5. Best regards, hope to hear from you soon
  6.  

Reviewer 3 Report

All text should be re-evaluated in terms of spelling and writing rules and fonts.
The membrane groups should be mentioned in abstract section.
Reference should be made to the established experiment on bacterial growth. In the discussion part, it should be discussed in more detail about why these two bacteria were selected and about the in vitro experimentation. Discussion should be expanded.

Author Response

Dear Reviewer:

Thank you for your remarks regarding our manuscript entitled “Bacterial growth on three non-resorbable polytetrafluoroethylene (PTFE) membranes. An in vitro study”.

Grammatical and spelling were revisited throughout the article.

We elaborated on the types of the membranes in the abstract section (lines 13 – 14).

We also, as requested, elaborated on the species of  the bacteria tested in the discussion (lines 243 – 248).

Best Regards.

Round 2

Reviewer 1 Report

Dear authors, 

The authors improved substantially the manuscript with the inclusion of characterization details, discussion and correction in the conclusion section. 

Moreover, applying more than one bacteria strain. 

My small comment in this second review is: I suggest the authors to remove the commercial names of the membranes in the abstract. The names should be exposed with all these details in the Materials and Methods section and not in the abstract. 

Author Response

Dear reviewer, 

We want to thank you for your time and efforts to revise our manuscript. Your comments  improved the quality of our study. 

The brand names of the tested membranes were removed from the abstract and are mentioned in the Materials and Methods section, as you suggested.

Best Regards.

Reviewer 2 Report

Since the manuscript has been heavily revised, I would like to reverse my original decision and suggest that it is acceptable.

Author Response

Dear reviewer, 

We want to thank you for your time and efforts to revise our manuscript. Your comments  improved the quality of our study.

Best Regards.

Reviewer 3 Report

.

Author Response

(The authors gave the same response as above.)
